# ICLR: Iterative Optimization for Information Extraction on In-Context Learning via Rule Filtering

## Abstract

Existing information extraction (IE) tasks, such as named entity recognition (NER) and relation extraction (RE), typically rely on fine-tuning or few-shot learning methods. In few-shot learning, large language models (LLMs) demonstrate excellent performance through in-context learning (ICL), which involves guiding the model by providing a few examples or rules in the prompt. However, a major challenge with this approach is the selection and optimization of contextual information for diverse IE tasks. In this work, we introduce ICLR (Iterative Context Learning Rule), a control-theoretic framework that models rule optimization as an adaptive filtering problem for comprehensive information extraction. We treat rules as controllable state variables and design an observer system to monitor and control LLM behavior indirectly, without modifying model parameters. Our method iteratively estimates and updates the optimal rule combinations using performance feedback, thereby reformulating the traditionally complex problem of LLM control into a well-defined state-space optimization that generalizes across multiple IE tasks. We evaluate ICLR on both NER datasets (CoNLL03, ACE05, GENIA) and RE datasets (NYT, CoNLL04), demonstrating rapid convergence and superior performance with minimal training data requirements. Our approach achieves up to 10% performance improvement over state-of-the-art ICL methods while requiring no additional model training and ICLR provides the first control-theoretic foundation for understanding and optimizing in-context learning behavior in information extraction.

## 1 Introduction

Recent information extraction (IE) methods, such as named entity recognition (NER) and relation extraction (RE), increasingly rely on in-context learning (ICL), where large language models (LLMs) (Brown et al., 2020) are guided by examples. While it's commonly thought that LLMs learn from these specific examples, evidence suggests a different mechanism. Min et al. (2022) and Huang et al. (2025) have demonstrated that models perform well even with incorrect labels, suggesting that LLMs are more responsive to abstract structural patterns than to concrete input-output pairs. For IE tasks specifically, this phenomenon is particularly pronounced due to inherent inconsistencies in annotation definitions across datasets. This observation leads to a fundamental question: what drives effective in-context learning for tasks like information extraction?

Recent work, particularly GuideNER (Huang et al., 2025), provides compelling evidence that explicit annotation guidelines outperform traditional example-based approaches in NER tasks. However, GuideNER's approach of selecting top-k rules based on simple frequency heuristics has inherent limitations in capturing the complex interdependencies between rules and their collective impact on performance across diverse IE tasks. Figure 1 illustrates this limitation: GuideNER requires traversing the complete dataset to identify the top-k most frequent rules, which is highly resource-intensive. In contrast, our method only needs to utilize a small portion of the dataset to automatically search for optimal particle combination strategies, thereby achieving superior performance. Moreover, by abstracting rules into subcategories, our method can generalize across all information extraction tasks.

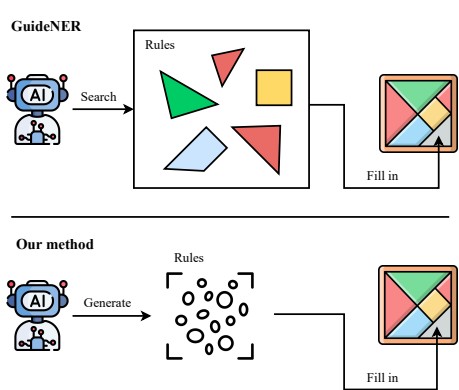

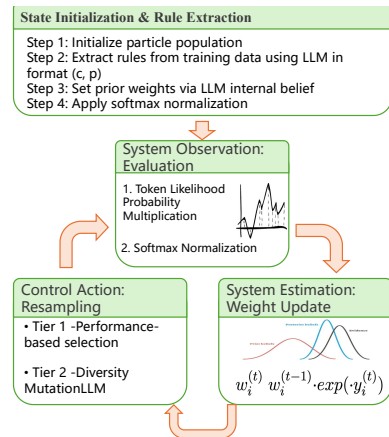

Figure 1: Comparison between GuideNER's static rule selection and our dynamic optimization framework

Figure 2: ICLR algorithm overview showing the complete iterative process from initialization to resampling

Building on this insight, we propose to view rules not merely as contextual information, but as **controllable state variables** that define the LLM's operational behavior for information extraction. Traditional approaches to LLM optimization face a fundamental challenge: the massive parameter space renders direct observation and control intractable. However, by treating rules as externally observable and controllable state variables, we can construct an observer system that enables systematic monitoring and control of LLM behavior without parameter modification.

This reformulation transforms the traditionally intractable problem of LLM control into a well-defined state-space optimization problem. Unlike internal LLM parameters, rules possess advantageous properties for control: external observability, semantic interpretability, systematic controllability, and lower dimensionality.

In this paper, we introduce **ICLR (Iterative Context Learning Rule)**, a control-theoretic framework that treats rule optimization as an adaptive filtering problem for comprehensive information extraction. Our approach iteratively estimates and updates the optimal rule state through performance feedback, creating a closed-loop system where observed IE performance directly influences subsequent rule modifications.

Our key contributions are threefold:

- We propose the first control-theoretic framework for ICL optimization in information extraction, providing a new theoretical foundation for understanding LLM behavior through external state variables;

- We develop ICLR, an efficient adaptive filtering algorithm that demonstrates both rapid convergence and data efficiency, extracting optimal rule combinations with minimal training samples while achieving faster optimization compared to existing methods;

- We demonstrate consistent improvements over state-of-the-art methods across both NER and RE datasets while requiring no additional model training and significantly shorter prompts.

## 2 RELATED WORK

### 2.1 IN-CONTEXT LEARNING FOR INFORMATION EXTRACTION

In-context learning has emerged as a powerful paradigm for natural language processing tasks, allowing large language models to adapt to new tasks through demonstration examples without parameter updates (Brown et al., 2020; Wei et al., 2022; Min et al., 2022). Traditional ICL approaches for information extraction rely on example-based demonstrations, where models learn input-output

mappings through pattern recognition from the provided examples (Dong et al., 2022; Li et al., 2023b).

However, example-based ICL faces challenges due to inconsistent annotation definitions across datasets, making it difficult for models to learn consistent patterns from conflicting examples. Huang et al. (2025) proposed GuideNER, which replaces example-based demonstrations with LLM-generated annotation guidelines, showing that explicit rules outperform traditional example-based methods. However, GuideNER's extension to comprehensive information extraction tasks remains underexplored, and it relies on heuristic frequency-based filtering rather than systematic optimization strategies for rule combination across diverse IE tasks.

For relation extraction specifically, ICL approaches face additional challenges in capturing complex inter-entity dependencies and contextual relationship patterns. GPT-RE retrieves task-aware demonstrations and augments them with label-guided reasoning to boost sentence-level RE, yet performance remains sensitive to example quality and handling of the no-relation class (Wan et al., 2023). Wadhwa et al. (2023) show that few-shot LLMs can approach fully supervised baselines, but they also document substantial variance across prompts and evaluation protocols. To better match structured outputs, CodeIE reformulates IE as code generation and leverages code LLMs for schema-constrained predictions, but still relies on fixed label verbalizations and inherits ICL's sensitivity to example selection (Li et al., 2023a). The need for structured rule representations that can handle both entity recognition and relation identification simultaneously presents unique optimization challenges that current methods do not adequately address.

### 2.2 OPTIMIZATION APPROACHES FOR LLM BEHAVIOR

The challenge of optimizing LLM behavior has been approached from various angles, which can be broadly categorized into three main directions: prompt engineering (Liu et al., 2023), fine-tuning methods (Wei et al., 2021), and retrieval-augmented approaches (Liu et al., 2021). Prompt engineering focuses on crafting optimal input formulations to elicit desired model behaviors without modifying model parameters. Fine-tuning approaches adapt pre-trained models to specific tasks through continued training, though this requires substantial computational resources and labeled data. Retrieval-augmented methods enhance model performance by dynamically incorporating relevant external information during inference.

Most existing methods focus on improving demonstration selection through similarity-based retrieval (Rubin et al., 2021) or supervised learning approaches (Li et al., 2023b). These approaches typically rely on semantic similarity metrics or learned representations to identify the most relevant examples for a given input. However, such methods often fail to account for the complex interactions between multiple examples and their collective impact on model behavior.

However, these approaches face a fundamental limitation: the massive parameter space of LLMs renders direct observation and control intractable, leading to heuristic optimization strategies that lack systematic principles.

Recent work has explored population-based optimization strategies for LLM reasoning. Qi et al. (2024) introduced evolutionary algorithms for multi-modal reasoning, using genetic operators to balance quality and diversity in solution generation. While their approach demonstrates the potential of population-based methods, it focuses on reasoning path optimization rather than systematic rule optimization for structured prediction tasks like NER.

Despite these advances, existing optimization approaches remain limited by the "black box" nature of LLMs. Current methods either require extensive computational resources for parameter modification or rely on heuristic strategies without theoretical foundations. The application of systematic control frameworks to LLM optimization, particularly for ICL rule selection, remains largely unexplored, presenting opportunities for more principled optimization approaches that treat rules as external state variables.

## 3 METHODOLOGY: ICLR

Our ICLR framework consists of a comprehensive control-theoretic approach for rule optimization, as illustrated in Figure 2 and Figure 3. Figure 2 shows the algorithmic overview with the

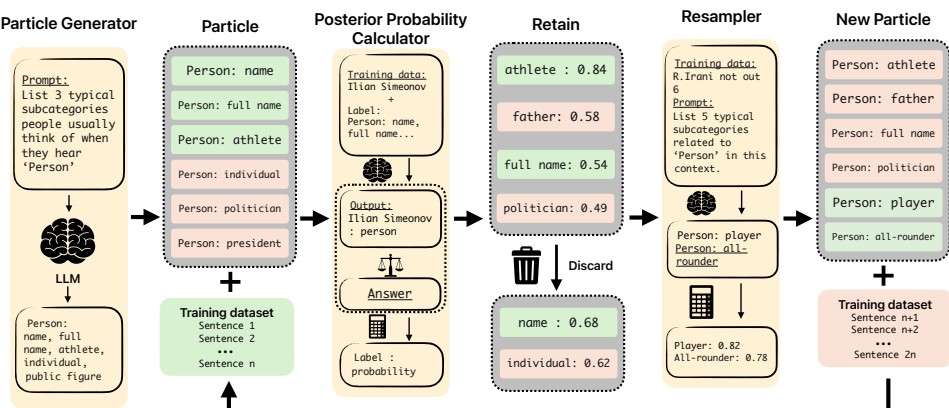

Figure 3: Overall framework of ICLR showing the particle-based rule optimization pipeline with iterative generation (Particle Generator), evaluation(Posterior Probability Calculator), selection (Retain), and mutation (Resampler) phases guided by LLM performance feedback.

four key steps of our adaptive filtering process, while Figure 3 presents the overall system architecture demonstrating the particle-based optimization flow. Figure 3 illustrates the particle-based rule optimization process. First, in the Particle Generator stage, the LLM generates several typical subcategories related to the target concept (e.g., Person), which, together with their labels, form a batch of "particles." The red particles in the figure represent those retained from the previous round, while the green particles represent the newly generated ones in the current round. Next, in the Posterior Probability Calculator stage, the model matches these particles with samples from the training dataset and computes their confidence scores on the current batch, thereby obtaining the posterior probabilities of each particle. In the Retain stage, high-confidence particles are preserved, while low-confidence ones are discarded. To maintain diversity, the process proceeds to the Resampler stage, where new or mutated particles are generated based on new training data and prompts to supplement and expand the particle set. Finally, in the New Particle stage, the newly generated and retained particles are combined and fed into the next batch of training data for further updating and evaluation. Through this iterative process, the particle set is progressively optimized, resulting in a more robust and efficient rule collection.

## 3.1 PROBLEM FORMULATION

Given a pre-trained large language model $\mathcal{M}$ and a target dataset $\mathcal{D} = \{(x_i, y_i)\}_{i=1}^{N}$ with training split $\mathcal{D}_{train}$ and test split $\mathcal{D}_{test}$, our goal is to find an optimal rule list $\mathcal{R}^*$ that maximizes the model's information extraction performance on the test set.

### 3.1.1 DATASET-SPECIFIC OPTIMIZATION

Since different datasets have varying data distributions and annotation conventions, we need to adapt the rule selection to each specific dataset. Using only the training portion $\mathcal{D}_{train}$, we extract and optimize a rule list $\mathcal{R}^*$ that is tailored to both the dataset characteristics and the specific LLM's behavior.

### 3.1.2 ICL-BASED EXTRACTION

For any input $x$, the extraction process follows:

$$\hat{y} = \mathcal{M}(\text{Prompt}(x, R_t)) \tag{1}$$

where $\mathcal{M}$ is the pre-trained LLM, $\text{Prompt}(x, R_t)$ constructs the input prompt by combining text $x$ with the optimal rule configuration $R_t$, and $\hat{y}$ is the predicted extraction output.

### 3.1.3 OPTIMIZATION OBJECTIVE

We seek to find the optimal rule configuration that maximizes performance on unseen data. Following standard machine learning practice, we split the available training data into training and validation sets, and optimize rules based on validation performance:

$$R_t^* = \arg\max_{R_t} \frac{1}{|\mathcal{D}_{\text{val}}|} \sum_{(x,y)\in\mathcal{D}_{\text{val}}} F(\mathcal{M}(\text{Prompt}(x, R_t)), y) \tag{2}$$

where $R_t = \{(p_i^{(t)}, c_i, w_i^{(t)})\}_{i=1}^N$ represents a rule configuration with $N$ particles at iteration $t$, where $p_i^{(t)}$ is the $i$-th subcategory pattern (particle), $c_i$ is its corresponding entity label, and $w_i^{(t)}$ is the confidence weight, $F(\cdot, \cdot)$ is a performance metric (e.g., F1 score). The rule extraction and initial population are derived from $\mathcal{D}_{\text{train}}$, while the optimization objective is evaluated on the held-out validation set $\mathcal{D}_{\text{val}}$ to prevent overfitting. Final evaluation is performed on $\mathcal{D}_{\text{test}}$.

### 3.1.4 CHALLENGES WITH EXISTING APPROACHES

While existing rule-based ICL methods like GuideNER have demonstrated the effectiveness of annotation guidelines over examples, they lack systematic optimization strategies for rule selection and combination. Current approaches rely on heuristic frequency-based filtering, which fails to capture the complex interdependencies between rules and their collective impact on IE performance.

### 3.1.5 CONTROL-THEORETIC REFORMULATION

Traditional LLM optimization faces a fundamental challenge: the massive parameter space (billions of parameters) renders direct observation and control intractable. We address this by treating rules as a low-dimensional, observable interface to the LLM system.

Recent findings suggest that in-context learning operates as a rule-based inference system, where the quality of rules—not examples—determines performance. This motivates reformulating the optimization as a control system problem, where rules serve as externally controllable state variables:

$$R_{t+1} = f(R_t, u_t) \quad \text{(Rule Evolution)} \tag{3}$$
$$y_t = h(R_t) \quad \text{(Performance Observation)} \tag{4}$$

where $R_t$ represents the rule configuration at iteration $t$, $u_t$ denotes control actions (rule modifications), and $y_t$ is the observed IE performance.

## 3.2 ADAPTIVE RULE FILTERING ALGORITHM

The discrete, combinatorial nature of rule spaces and nonlinear performance mappings makes classical control methods inappropriate. We develop an adaptive filtering approach that iteratively estimates optimal rule configurations through performance feedback.

As illustrated in Figure 2, our approach follows a systematic particle filtering process:

### 3.2.1 PARTICLE-BASED STATE REPRESENTATION

In our framework, each **particle** represents a subcategory pattern (e.g., "person names with academic titles"), and each **rule** consists of a particle paired with its corresponding entity label (e.g., "person names with titles" → PER).

At time step $t$, the complete rule configuration is:

$$R_t = \{(p_i^{(t)}, c_i, w_i^{(t)})\}_{i=1}^N \tag{5}$$

as defined in Equation (2), where each tuple represents a complete rule with its associated weight.

Our adaptive filtering approach optimizes the mapping relationships by filtering out noisy or ineffective subcategory patterns while retaining those that contribute most to extraction performance. The

"filtering" terminology reflects the need to reduce noise in particle weights and identify the most reliable pattern-label mappings.

The final output combines the best-performing particles from the evolved population to form an optimized rule list containing multiple complementary rules across different entity types.

The filtering process consists of four steps:

**0. Initialization (Rule Extraction):** We initialize the particle population by extracting initial rules from the training dataset using the GuideNER approach. For each input-label pair $(x_j, y_j)$ in the training set, we use LLM to summarize rule patterns:

$$p_i^{(0)} = \text{LLM}_{\text{extract}}(x_j, y_j, \text{prompt}_{\text{summary}}) \tag{6}$$

The initial particles are assigned prior weights based on their alignment with LLM's internal rule representations:

$$w_i^{(0)} = \text{softmax}(\text{LLM}_{\text{confidence}}(p_i^{(0)}, \theta)) \tag{7}$$

where $\text{LLM}_{\text{confidence}}(p_i^{(0)}, \theta)$ represents the average logits of the LLM's output when using rule $p_i^{(0)}$ as a guideline, reflecting the model's confidence in the rule's effectiveness.

**1. Evaluation (ICL-based Observation):** Each particle is evaluated through ICL-based IE inference on a validation subset. The LLM uses the complete rule (particle + label) as an annotation guideline to perform information extraction:

$$y_i^{(t)} = \text{softmax}\left(\prod_{j=1}^{|T_i|} P(t_j|\text{context}, (p_i^{(t)}, c_i))\right) \tag{8}$$

where $P(t_j|\text{context}, (p_i^{(t)}, c_i))$ represents the probability of generating token $t_j$ in the ICL-based IE output sequence $T_i$ when using rule $(p_i^{(t)}, c_i)$ as the guideline. This likelihood-based evaluation serves as the observation step in our control system, providing probabilistic performance feedback for each rule hypothesis.

**2. Weight Update (Bayesian Posterior):** Particle weights are updated using Bayesian inference, combining prior knowledge with observed performance:

$$w_i^{(t)} \propto w_i^{(t-1)} \cdot p(y_i^{(t)}|p_i^{(t)}) = w_i^{(t-1)} \cdot \exp(\beta \cdot y_i^{(t)}) \tag{9}$$

where $w_i^{(t-1)}$ represents the prior weight and $p(y_i^{(t)}|p_i^{(t)})$ is the likelihood of observing performance $y_i^{(t)}$ given rule $p_i^{(t)}$. The parameter $\beta$ controls the selection pressure in the likelihood function.

**3. Multi-level Resampling with Rule Mutation:** We employ an ICL-based resampling strategy that includes rule generation and mutation, adapting the LLM-guided mutation approach from Qi et al. (2024):

- **Tier 1 (Performance-based Selection):** Remove particles with weights below threshold based on ICL evaluation results
- **Tier 2 (Diversity Mutation):** Apply semantic mutations to existing particles using ICL prompting. Each selected particle undergoes controlled semantic mutation through LLM-guided rule generation:

$$p_i^{(t|t-1)} = \text{LLM}_{\text{mutate}}(p_i^{(t-1)}, u_t) \tag{10}$$

Three mutation strategies are employed:
  - **Refinement:** Increase specificity
  - **Generalization:** Increase coverage
  - **Contextualization:** Generate domain-specific rule variants

**Component Assignment for New Particles:** For newly generated or mutated particles, the three components are obtained as follows:

- **Pattern** ($p_i^{(t)}$)**:** Generated through LLM mutation as shown in Equation (10)

- **Label** ($c_i^{(t)}$)**:** Inherited from the parent particle to maintain semantic consistency

- **Weight** ($w_i^{(t)}$)**:** Assigned based on perplexity under the LLM:

$$w_i^{(t)} = \text{softmax}(-\text{PPL}(p_i^{(t)})) \tag{11}$$

where $\text{PPL}(p_i^{(t)}) = \exp(-\frac{1}{|p_i^{(t)}|} \sum_j \log P(t_j | t_{<j}))$ is the perplexity of particle $p_i^{(t)}$ computed as the exponentiated negative log-likelihood per token. Lower perplexity indicates that the rule text is more consistent with the model's learned patterns. This integrated resampling process ensures that the particle population evolves systematically while maintaining the diversity necessary for effective exploration of the rule space.

## 4 EXPERIMENT

We conduct extensive experiments on multiple Information Extraction tasks to evaluate the effectiveness of our proposed approach. For fairness, we follow previous work and adopt F1-score as the evaluation metric throughout our experimental analysis. Our comprehensive evaluation demonstrates the superior performance of our method across diverse IE datasets, including NER and RE, and validates the generalization of our approach.

### 4.1 DATASETS

For named entity recognition, we evaluate on three datasets: CoNLL-2003(Tjong Kim Sang & De Meulder, 2003) is a standard benchmark containing news articles from the Reuters corpus, with four entity types: Person (PER), Location (LOC), Organization (ORG), and Miscellaneous (MISC). While ACE 2005(Walker et al., 2006) focuses on entity detection and classification in news and conversational data. And GENIA(Kim et al., 2003) is a biomedical NER dataset containing abstracts from the MEDLINE database.

For relation extraction, we utilize two established benchmarks: NYT(Riedel et al., 2010) (New York Times) dataset, which contains news articles with distant supervision for relation extraction between entity pairs. And CoNLL04(Roth & Yih, 2004) is a manually annotated dataset that provides both entity and relation annotations.

This diverse collection of datasets allows us to demonstrate the effectiveness of our approach across different domains, including news, biomedical, general text, as well as different task formulations, thereby providing comprehensive evidence of our method's robustness and generalizability.

### 4.2 EXPERIMENTS SETUP

All experiments are conducted on a computing cluster equipped with H100 GPUs for computational acceleration. They are implemented using PyTorch 2.6.0 and transformers 4.51.3. We employ four foundation models with varying scales, languages, and training datasets: Qwen2.5-3B, Qwen2.5-7B, Llama3.1-8B and Pixtral-12B. This selection allows us to investigate the impact of different model size (3B, 7B, 8B and 12B parameters) on our method's performance. All models are tested with temperature set to 0.0 and random seed fixed at 42 to ensure reproducibility of the experiments and stability when generating posterior probabilities. In addition, during the computation of prior probabilities, all models are evaluated in eval mode to disable dropout and other stochastic behaviors. This configuration ensures consistent experimental conditions and enables direct performance comparison across different models and datasets.

### 4.3 BASELINES

For baseline comparisons, we adopt the most widely-used ICL approaches in IE domain. This means that we do not employ any fine-tuned models or models with built-in chain-of-thought capabilities (e.g., Qwen3). We rely entirely on the models' internal information extraction abilities to learn from

context. Our baseline methods include: GuideNER(Huang et al., 2025), the current state-of-the-art method, assists in-context learning by enabling the model to learn from the internal relationships within examples. However, GuideNER is specifically designed for NER tasks only, so in the following comparisons, we only evaluate GuideNER's performance on NER datasets. Therefore, in subsequent comparison results on non-NER tasks, GuideNER is represented with "—" to indicate its inapplicability. The other baseline is CodeIE(Li et al., 2023a), which employs a sophisticated example selection process to enhance model performance. It is a strong baseline, featuring a carefully designed example selection strategy that is model-agnostic and consistently improves performance across different architectures and tasks. Additionally, we test the direct input-output (IO) approach, where the dataset is run on the original model without any examples or demonstrations, serving as the most fundamental baseline.

This experimental design ensures that all baseline methods operate under the same paradigm of pure in-context learning, enabling fair comparison of different approaches' effectiveness.

| Method | Model | NER | | | RE | | Token Cost |
|--------|-------|--------|-------|-------|------|---------|------------|
| | | CoNLL03 | ACE05 | GENIA | NYT | CoNLL04 | |
| IO | Qwen-2.5-3b | 60.29 | 20.43 | 30.43 | 7.00 | 3.05 | |
| | Qwen-2.5-7b | 62.37 | 36.53 | 45.20 | 10.62 | 4.32 | |
| | Llama-3.1-8b | 47.67 | 36.49 | 31.79 | 11.05 | 4.10 | 285 |
| | Pixtral-12B | 31.98 | 25.51 | 14.02 | 10.47 | 0.00 | |
| CodeIE | Qwen-2.5-3b | 61.26 | 25.52 | 40.73 | 10.38 | 5.59 | |
| | Qwen-2.5-7b | 65.32 | 42.43 | 47.59 | 15.43 | $10.56^{*}$ | |
| | Llama-3.1-8b | 56.20 | 40.98 | 41.26 | 14.66 | 9.20 | 1172 |
| | Pixtral-12B | 34.28 | 26.54 | 16.53 | 15.12 | 5.43 | |
| GuideNER | Qwen-2.5-3b | 63.32 | 27.43 | 41.49 | — | — | |
| | Qwen-2.5-7b | 65.10 | 41.57 | 47.43 | — | — | |
| | Llama-3.1-8b | 61.38 | 44.97 | 42.86 | — | — | 506 |
| | Pixtral-12B | 34.76 | 27.64 | 18.03 | — | — | |
| ICLR | Qwen-2.5-3b | 65.12 | 35.46 | 46.98 | 11.47 | 4.42 | |
| | Qwen-2.5-7b | 72.83 | 46.94 | 51.36 | 15.47 | 8.46 | |
| | Llama-3.1-8b | 69.14 | 53.10 | 50.15 | 15.18 | 8.93 | 501 |
| | Pixtral-12B | 45.54 | 32.87 | 30.65 | 19.6 | 5.75 | |

Table 1: Performance comparison of different methods (IO, CodeIE, GuideNER, and ICLR) across Qwen-2.5, Llama-3.1, and Pixtral models on NER (CoNLL03, ACE05, GENIA) and RE (NYT, CoNLL04) benchmarks. The best result for each dataset is highlighted in bold with $^{*}$. Gray shading indicates significant test p-value smaller than 0.05 when compared with IO, CodeIE and GuideNER.

## 4.4 MAIN RESULTS

Our experimental results, shown in Table 1, reveal several key findings that provide important insights into the effectiveness and limitations of in-context learning for information extraction tasks.

We observe that current ICL effectiveness remains constrained by models' inherent capabilities, though parameter count does not necessarily correlate with performance. Despite Pixtral-12B having more parameters than Qwen-2.5-7B, it consistently underperforms across all tasks, suggesting that training data quality and architectural design may be more critical than raw parameter scale.

What's more, our method demonstrates substantial performance improvements over current state-of-the-art ICL approaches across all experimental tasks. ICLR achieves consistent gains ranging from 7.7% to 18.1% relative improvement on NER tasks, with particularly notable results on CoNLL03 where we achieve 72.83 F1 score compared to GuideNER's 65.10 using Qwen2.5-7B. For relation extraction tasks, our method shows even more pronounced advantages, delivering up to 87% relative improvement on challenging datasets like NYT with Pixtral-12B.

From a computational efficiency perspective, ICLR achieves comparable token consumption to GuideNER (501 vs 506 tokens) while delivering superior performance, and remains 57% more efficient than CodeIE (1172 tokens).

Finally, our approach demonstrates superior generalizability compared to current state-of-the-art methods. While GuideNER is specifically designed and limited to NER tasks, our method successfully extends to comprehensive information extraction scenarios, including both named entity recognition and relation extraction tasks. As evidenced in our experimental results, ICLR achieves consistent performance improvements across diverse IE tasks, enabling seamless adaptation across different information extraction paradigms. For a complete understanding of the operational mechanics of our method, detailed execution steps of all components are provided in Appendix A.

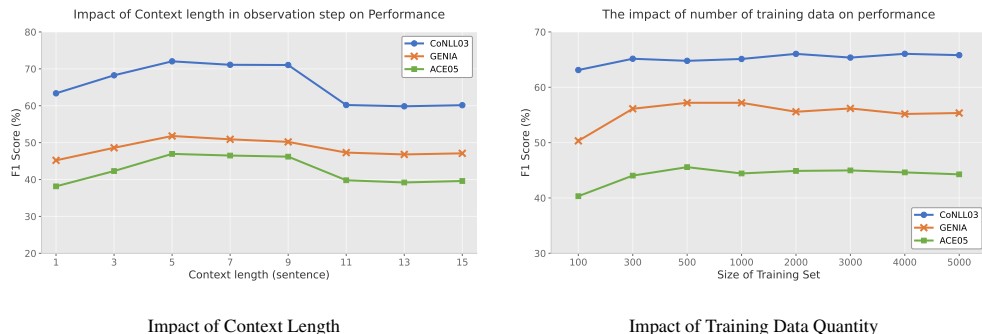

Impact of Context Length                                       Impact of Training Data Quantity

Figure 4: Performance Analysis: Token consumption and performance under different conditions.

### 4.5 PARAMETER SENSITIVITY ANALYSIS

Since ICLR functions as a unified sequential framework with tightly coupled components that preclude conventional ablation studies, we perform extensive parameter analysis on two key factors: context window length and training data size.

### 4.5.1 IMPACT OF CONTEXT WINDOW LENGTH ON ICLR PERFORMANCE

The context window length determines how much historical information our method considers when computing posterior probabilities at each observation step. Figure 4 shows that performance improves significantly from single-sentence to four-sentence contexts due to smoother joint likelihood functions that provide more stable gradients. However, performance declines beyond 9 sentences as excessive observations lead to overly averaged particle weights, preventing effective updates.

### 4.5.2 IMPACT OF AMOUNT OF TRAINING DATA QUANTITY ON ICLR PERFORMANCE

Our experimental results demonstrate distinct performance saturation patterns across different datasets. As shown in Figure 4, our method exhibits rapid performance convergence with relatively small training sets. Specifically, on CoNLL03, performance reaches approximately 63% F1 score with just 100 training samples and plateaus around 65-66% with 300+ samples. Similarly, GENIA achieves 50% F1 with 100 samples and stabilizes around 55-57% with larger training sets. ACE05 shows a more gradual improvement from 40% to 45% F1 score. Notably, performance gains become marginal beyond 1000 training samples across all datasets, suggesting our method can achieve near-optimal performance with limited supervision, making it particularly suitable for resource-constrained scenarios.

## 5 CONCLUSION

In this paper, we propose ICLR, a general framework that can efficiently traverse training sets and rapidly extract specific relationships from training data using particle-based methods. We have conducted extensive experiments to demonstrate the effectiveness of our approach. Additionally, we acknowledge the limitations of our method, such as the generation of a large number of particles to achieve rapid convergence, which increases inference burden and computational costs. Therefore, our future research will focus on improving particle utilization efficiency to enhance the overall system performance.

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

## A    CASE STUDY

We present a case study using a CoNLL-2003 sentence to illustrate our three-stage workflow: particle generator, posterior probability calculator, and resampler. Here, $x$ represents label sequences, $N$ denotes the number of generation rules, and $Y$ represents input text. It is worth noting that the following case studies present simplified examples to illustrate our framework's workflow. All experiments follow the technical specifications detailed in Section 3.

### A.1    CASE STUDY 1: PARTICLE GENERATOR

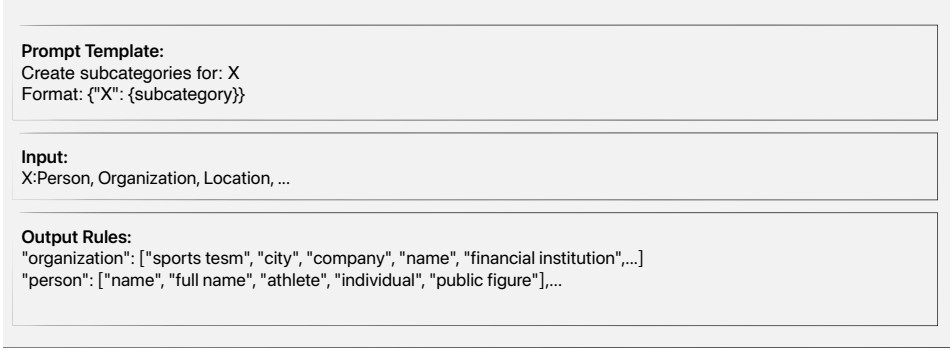

**Prompt Template:**
Create subcategories for: X
Format: {"X": {subcategory}}

**Input:**
X:Person, Organization, Location, ...

**Output Rules:**
"organization": ["sports tesm", "city", "company", "name", "financial institution",...]
"person": ["name", "full name", "athlete", "individual", "public figure"],...

Figure 5: Case study showing the prompt template for subcategory generation. The template guides the model to generate semantic subcategories for named entity labels, demonstrated with organization and person entity types.

Particle generation serves as the initial step in our framework to obtain initial particles and compute their corresponding prior probabilities. We demonstrate this process using the label "organization" as an example. As shown in Figure5, our prompt template instructs the model to act as a subcategory generation expert, decomposing broad entity labels into semantically distinct subcategories with associated probability weights. For the "organization" label, the system generates diverse subcategories such as "sports team", "city", "company", "name", and "financial institution". Each subcategory represents a different semantic interpretation of how organizations might appear in text, with the probability weights serving as prior probabilities that reflect their typical occurrence frequency.

### A.2    CASE STUDY 2: POSTERIOR PROBABILITY CALCULATOR

Following particle generation, we compute posterior probabilities by evaluating how well each generated rule performs on the actual classification task. Figure6 illustrates this process using the input text "EU rejects German call to boycott British lamb." The system applies the previously generated subcategory rules to identify entities in the given text. The likelihood of each particle is calculated

**Prompt Template:**
Classify entities in the text into predefined categories.
Label: Person, Organization, Location, ...
Input: Y
Output: Structured entity–category pairs

**Input Text:**
Eu rejects German call to boycott British lamb.

**Model Output Rules:**
"organization": ["sports team", "city", "company", "name", "financial institution",...]
"person": ["name", "full name", "athlete", "individual", "public figure",...]

**Model Output Rules and Calculated Posterior Probability:**
"organization": ["sports team: 0.9277", "city: 0.7073", "company: 0.4751", "name: 0.7891", "financial institution: 0.6779",...]
"person": ["name: 0.6850", "full name: 0.5437", "athlete: 0.1327", "individual: 0.6227", "public figure: 0.7104",...]

Figure 6: Case study demonstrating posterior probability calculation through likelihood estimation. The model evaluates generated rules against input text to compute particle posterior probabilities based on classification accuracy.

based on the model's ability to correctly identify and classify entities according to the generated rules. The posterior probability calculation applies Bayes' theorem to update particle weights based on how well each rule matches the observed entity patterns in the input text. For instance, the "organization" subcategories show varying performance scores: "sports team: 0.9277", "city: 0.7073", "company: 0.4751", demonstrating how different semantic interpretations receive different posterior weights based on their effectiveness. This likelihood-based evaluation ensures that particles with better classification performance receive higher posterior probabilities, enabling the system to focus on the most promising labeling hypotheses for subsequent resampling.

### A.3 CASE STUDY 3: RESAMPLER

The final stage employs an LLM-based resampling mechanism that performs context-aware particle mutation. As shown in Figure 7, given the input text "He said further scientific study was required and if it was found that action was needed it should be taken by the European Union," the system generates contextually relevant subcategories. The resampling process works as follows: the LLM analyzes the specific context and mutates the original particles to better fit the observed text patterns. For the "organization" label, the system generates context-specific subcategories such as "educational institution," "non-governmental organization," "research institution," and "financial institution," which are more relevant to the scientific and policy context of the input. Prior probabilities are computed to indicate higher prior probability. For instance, "educational institution: 0.340369" and "non-governmental organization: 0.297805" receive relatively high priors due to their contextual relevance. This perplexity-based weighting ensures that contextually appropriate mutations are favored during the resampling process.

## B USE OF LLMS

In this paper, we employed Large Language Models (LLMs) as core components of our proposed ICLR framework in three specific stages. First, we utilized LLMs for rule extraction and particle generation to create initial subcategory patterns from training data, ensuring systematic rule discovery. Second, we leveraged LLMs for performance evaluation through in-context learning inference, where models assess rule effectiveness on validation datasets. Third, we employed LLMs for rule mutation and resampling to generate semantically diverse rule variants during the optimization process.

**Prompt Template:**
Classify the label into more specific subcategories based on the provided context.

Label: X
Context Text: Y
Output: List of relevant subcategories

**Input Text:**
He said further scientific study was required and if it was found that action was needed it should be taken by the European Union .

**Model output:**
"organization": ["educational_institution", "non_governmental_organization", "research_institution", "financial_institution", "sports_team",...]
"person": ["celebrity", "family_member", "colleague", "fictional_character", "historical_figure",...]

**Prior Probability:**
"organization": ["educational_institution: 0.340369", "non_governmental_organization: 0.297805", "research_institution: 0.144011", "financial_institution: 0.141820", "sports_team: 0.046757",...]
"person": ["celebrity: 0.258245", "family_member: 0.060173", "colleague: 0.036731", "fictional_character: 0.370841", "historical_figure: 0.251535",...]

Figure 7: Case study demonstrating the resampling stage with particle mutation.

Additionally, we used LLMs to assist with writing and language improvement throughout the manuscript preparation process. This included grammar checking, sentence structure optimization, and clarity enhancement to improve the overall readability of our work. However, all core research ideas, methodological contributions, experimental designs, and conclusions remain entirely our own intellectual work.

