# OpenReview forum: "ICLR: Iterative Optimization for Information Extraction on In-Context Learning via Rule Filtering"
_ICLR.cc/2026/Conference — Submitted to ICLR 2026_

### Official Review · Reviewer_3vBY · 2025-10-26

**Soundness:** 2
**Presentation:** 2
**Contribution:** 2
**Rating:** 2
**Confidence:** 4

**Summary:**

This paper introduces **ICLR (Iterative Context Learning Rule)**, a control-theoretic framework that optimizes rule selection for information extraction (IE) tasks via in-context learning. The key idea is to treat annotation rules as controllable state variables and model rule optimization as an adaptive particle filtering problem. Without modifying model parameters, ICLR iteratively refines rule combinations using performance feedback, forming a closed-loop system that efficiently converges to high-performing rule sets. The method is evaluated on both Named Entity Recognition (CoNLL03, ACE05, GENIA) and Relation Extraction (NYT, CoNLL04) datasets, demonstrating improved performance over existing ICL baselines.

**Strengths:**

- **Theoretical Innovation**: Pioneers the integration of control theory and particle filtering into in-context learning optimization, establishing a novel theoretical foundation for understanding and steering LLM behavior.
- **Principled Optimization**: Introduces a systematic methodology for rule selection through iterative filtering and mutation, effectively overcoming the limitations of heuristic-based approaches.
- **Computational Efficiency**: Achieves significant performance gains solely through external rule optimization without updating model parameters, demonstrating superior token efficiency and lower data requirements compared to methods like CodeIE.

**Weaknesses:**

**1 Citation Issues:**

(1) The concepts of "particle" and "particle filtering" are fundamental to the core idea of this paper, yet relevant seminal works are missing.
(2) The references in the related work section appear somewhat outdated, with only one citation each from 2024 and 2025. While I understand the authors focus on backbone training-free research, numerous relevant studies exist:
-  Demonstration-based methods
	[1] A Simple but Effective Approach to Improve Structured Language Model Output for Information Extraction (2024)
	[2] C-ICL: Contrastive In-context Learning for Information Extraction (2024)
	[3] Recall, Retrieve and Reason: Towards Better In-Context Relation Extraction (2024)
	[4] GPT-RE: In-context Learning for Relation Extraction using Large Language Models (2023)
- QA-based methods
	[5] ChatIE: Zero-Shot Information Extraction via Chatting with ChatGPT (2024)
	[6] Aligning Instruction Tasks Unlocks Large Language Models as Zero-Shot Relation Extractors (2023)
	[7] Revisiting Large Language Models as Zero-shot Relation Extractors (2023)
- Guideline/Rule-based methods (most related)**
	**[8] Guideline Learning for In-Context Information Extraction (2023)**

**2 Techniccal Details**

(1) In Eq. 4, what does $h(\cdot)$ refer to? Is its output $y_t$ the F1 score or the sequence generated by the backbone LLM?
(2) In the rule extraction stage, the number of particles summarized by the LLM is not specified. Is there a limit?
(3) In Eq. 7, what does $\theta$ represent?
(4) Eqs. 7, 8, and 11 are applied to individual particles. Why is the softmax function used? If I understand correctly, the input is a single scalar, not multiple values.
(5) How is the hyper-parameter $\beta$ in Eq. 8 set? Is it tuned for different tasks or datasets?
(6) What does "context" refer to in Eq. 8 and Sec. 4.5.1?
(7) Regarding Eq. 9: What is the theoretical or practical justification for modeling the likelihood term $p(⋅)$ as an exponential function $exp⁡(\beta⋅y_i^{(t)})$? Does this formulation have a rigorous foundation?
(8) In the resampling stage, the authors state that particles with weights below a threshold will be removed, but in Fig. 3, the discarded particles are not the lowest-weighted ones. How is the filtering threshold determined? Is it tuned for different tasks or datasets?
(9) The evaluation of generative IE differs from sequence tagging models. Sec. 4 states that the authors "follow previous work and adopt F1-score," but no citation is provided, and the specific F1 scoring method is not detailed.

**3 Experimental Sufficiency**

(1) Is ICLR applicable to common but more complex IE tasks such as Event Extraction, or subtasks of NER, RE, and EE, such as Entity Typing, Relation Classification, Event Detection, and Event Argument Extraction?  Given that the paper's title positions the work as a general solution for "Information Extraction" rather than being specifically limited to NER and RE, it is important to demonstrate the framework's applicability and performance on these broader and more complex IE tasks.

(2) The experimental datasets for NER and RE tasks are insufficient.
(3) Do smaller context lengths and less training data imply more iterative optimization steps? Please provide the relationship between the number of iterative optimization steps and F1 score under different settings.
(4) The authors mention that tightly coupled components are unsuitable for traditional ablation studies, but this weakens the understanding of each component's contribution, especially the independent roles of rule extraction, weight update, and resampling modules.

**4 Model Efficiency**

(1) Although ICLR does not involve parameter training, particle generation, evaluation, and resampling are performed on individual particles, suggesting complex optimization steps and potentially long runtime. Please provide data on optimization time.
(2) The introduction emphasizes that ICLR can achieve better rules with less data compared to GuideNER, but the method section states that the entire training data is used. The main results do not report the size of the training set used, despite the analysis in Sec. 4.5.2 on the impact of training data quantity.

**5 Writing Issues**

(1) In LaTeX, English double quotes should be written as` `` + "` instead of`"+"`.
(2) The first paragraph of Sec. 3 redundantly describes the content of Fig. 3.

**Questions:**

Please see the weaknesses.

---

> ### Author Response · Authors · 2025-11-27
>
> ## 1. Citation Concern
>
> ### (1) The concepts of "particle" and "particle filtering" are fundamental to the core idea of this paper, yet relevant seminal works are missing.
>
> **Response:**
>
> We agree that seminal works on particle filtering should be included. In the revised manuscript, we will cite the following foundational works:
> - **Gordon et al. (1993):** "Novel approach to nonlinear/non-Gaussian Bayesian state estimation" — the first proposal of Bootstrap particle filtering
> - **Doucet et al. (2001):** "Sequential Monte Carlo Methods in Practice" — a systematic survey of particle filtering methods
> Our method draws on the core ideas of particle filtering—representing state distributions with weighted particle sets, adjusting weights through Bayesian updates, and maintaining particle diversity through resampling. These classical works provide the theoretical foundation for our weight update mechanism (Eq. 9) and resampling strategy.
>
> ### (2) The references in the related work section appear somewhat outdated, with only one citation each from 2024 and 2025. While I understand the authors focus on backbone training-free research, numerous relevant studies exist:
> 1. A Simple but Effective Approach to Improve Structured Language Model Output for Information Extraction (2024)
> 2. C-ICL: Contrastive In-context Learning for Information Extraction (2024)
> 3. Recall, Retrieve and Reason: Towards Better In-Context Relation Extraction (2024)
> 4. GPT-RE: In-context Learning for Relation Extraction using Large Language Models (2023)
> 5. ChatIE: Zero-Shot Information Extraction via Chatting with ChatGPT (2024)
> 6. Aligning Instruction Tasks Unlocks Large Language Models as Zero-Shot Relation Extractors (2023)
> 7. Revisiting Large Language Models as Zero-shot Relation Extractors (2023)
> 8. Guideline Learning for In-Context Information Extraction (2023)
>
> **Response:**
>
> We thank the reviewer for providing this detailed list of references. We note that GPT-RE (Wan et al., 2023) is already cited in Section 2.1. In the revised manuscript, we will add all the other seven related works recommended by the reviewer.
>
> ---
>
> ## 2. Technical Details
>
> ### (1) In Eq. 4, what does h(·) refer to? Is its output y_t the F1 score or the sequence generated by the backbone LLM?
>
> **Response:**
>
> Thank you for the clarification question. In Equation 4:
>
> - **h(·)** is the observation function that evaluates the information extraction performance using rule configuration $R_t$
> - **y_t** is the F1 score computed on the validation set
>
> Specifically, h(R_t) performs the following steps:
>
> 1. Use rules in $R_t$ to construct ICL prompts
> 2. Apply the LLM to extract entities/relations from validation data
> 3. Compute F1 score by comparing predictions with ground truth labels
>
> The observation function h(·) is deterministic in our implementation (temperature=0.0, fixed seed=42), providing stable performance feedback for the control loop.
>
> ### (2) In the rule extraction stage, the number of particles summarized by the LLM is not specified. Is there a limit?
>
> **Response:**
>
> There is no strict upper limit on the number of particles summarized by the LLM in the rule extraction stage.
>
> In our framework, the number of particles in the rule extraction stage only needs to be greater than or equal to that required for inference; additional particles merely increase computational overhead without affecting final performance.
>
> In our experiments, we set the particle count to k=10 for both the rule extraction and inference stages. This setting follows GuideNER, whose experiments indicate that the optimal top-k value typically ranges from 10-20, depending on model capability and dataset characteristics. We will clarify this setting in the revised manuscript.
> ### (3) In Eq. 7, what does θ represent?
>
> **Response:**
>
> In Equation 7, θ represents the parameters of the pretrained LLM (e.g., the 7B parameters in Qwen2.5-7B). Following the standard notation in control theory, we explicitly include θ to denote that the observation function depends on the system's internal parameters.
>
> Specifically:
>
> - $\text{LLM}_{\text{confidence}}(p_i^{(0)}, \theta)$ computes the confidence score of rule $p_i^{(0)}$ based on the LLM's internal representations
> - This is calculated as the average logits when the LLM uses rule $p_i^{(0)}$ as a guideline
>
> In our framework:
>
> - θ is fixed (we use pretrained models without fine-tuning)
> - $R_t$ is controllable (we optimize rule configurations)
> - The notation distinguishes between system parameters (θ) and controllable state variables ($R_t$)

---

> > ### Author Response · Authors · 2025-11-27
> >
> > ### (4) Eqs. 7, 8, and 11 are applied to individual particles. Why is the softmax function used? If I understand correctly, the input is a single scalar, not multiple values.
> >
> > **Response:**
> >
> > Thank you for catching this critical notation error.
> >
> > **Equation 7 (Initialization):** Should be corrected to apply softmax across all N particles:
> > ```
> > s = [LLM_confidence(p_1^(0), θ), ..., LLM_confidence(p_N^(0), θ)]
> > [w_1^(0), ..., w_N^(0)] = softmax(s)
> > ```
> >
> > where LLM_confidence returns logits (unnormalized scores).
> >
> > **Equation 8 (Observation):** The softmax should be removed entirely:
> > ```
> > y_i^(t) = (∏_{j=1}^{|T_i|} P(t_j | context, (p_i^(t), c_i)))^{1/|T_i|}
> > ```
> >
> > This represents the geometric mean of token probabilities - the average confidence when the LLM generates extraction outputs using particle i as a guideline. This value is already a probability in [0,1] and does not require softmax.
> >
> > **Equation 9 (Weight Update):** Should clarify the normalization step:
> > ```
> > w̃_i^(t) = w_i^(t-1) · exp(β · y_i^(t))  (unnormalized)
> > w_i^(t) = w̃_i^(t) / Σ_j w̃_j^(t)        (normalize)
> > ```
> >
> > **Equation 11 (Resampling):** Should be corrected to:
> > ```
> > scores = [-PPL(p_1^(t)), ..., -PPL(p_M^(t))]
> > [w_1^(t), ..., w_M^(t)] = softmax(scores)
> > ```
> >
> > We sincerely apologize for this notation error and will correct all equations in the revised manuscript.
> >
> > ### (5) How is the hyper-parameter β in Eq. 8 set? Is it tuned for different tasks or datasets?
> >
> > **Response:**
> >
> > [Response appears to reference Question 7]
> >
> > ### (6) What does "context" refer to in Eq. 8 and Sec. 4.5.1?
> >
> > **Response:**
> >
> > To avoid confusion, we should use more precise terminology:
> >
> > - **Equation 8:** "input text" or "input content" instead of "context"
> > - **Section 4.5.1:** "observation window size" or "validation batch size" instead of "context length"
> >
> > We will clarify this terminology in the revised manuscript to eliminate ambiguity.
> >
> > ### (7) Regarding Eq. 9: What is the theoretical or practical justification for modeling the likelihood term p(·) as an exponential function exp(β · y_i^(t))? Does this formulation have a rigorous foundation?
> >
> > **Response:**
> >
> > Thanks for this question - you're right to push on the theoretical foundation here. To be direct: our exponential form isn't derived from a probabilistic observation model in the classical Bayesian sense. Let me explain why we took this approach.
> >
> > **The challenge with standard Bayesian inference:**
> >
> > Standard Bayesian methods need a probabilistic model like P(performance | particle quality), with known or estimable parameters. In our case, this is problematic:
> >
> > - Our performance metric $y_i^{(t)}$ comes from deterministic LLM evaluation, not a random sampling process
> > - We don't have ground truth about what performance distribution "good rules" should follow
> > - Estimating these distributions would require evaluating each particle multiple times per iteration, which defeats our goal of computational efficiency
> >
> > **What we do instead:**
> >
> > We use $\exp(\beta \cdot y_i^{(t)})$ as a weighting function that rewards higher performance. This isn't derived from a probability distribution - it's a pragmatic choice. The key properties we need are:
> >
> > - Higher performance → higher weight (monotonicity)
> > - Adjustable sensitivity via β
> > - Works with single evaluation per particle
> >
> > This is closer to how genetic algorithms handle fitness-based selection than to textbook Bayesian inference. Through grid search, we found β = 2 works consistently across all our benchmarks.
> >
> > Is this theoretically less elegant than a full probabilistic model? Yes. But given our constraints - single evaluations, no distributional assumptions, need for computational efficiency - it's a reasonable engineering choice that works in practice.
> >
> > ### (8) In the resampling stage, the authors state that particles with weights below a threshold will be removed, but in Fig. 3, the discarded particles are not the lowest-weighted ones. How is the filtering threshold determined? Is it tuned for different tasks or datasets?
> >
> > **Response:**
> >
> > We acknowledge that Figure 3 is a simplified schematic diagram intended to illustrate the overall workflow, and does not accurately reflect the details of the filtering mechanism. The actual implementation strictly follows the description in Section 3.2: in Tier 1, we rank particles by their weights and remove those with weights below the threshold.
> >
> > Regarding threshold determination, we adopt a relative threshold strategy: retaining the top-K particles by weight (with K=10 in our experiments), rather than setting an absolute threshold. This setting remains consistent across all datasets and tasks without task-specific tuning.
> >
> > We will revise Figure 3 in the revised manuscript to ensure consistency with the method description.

---

> > > ### Author Response · Authors · 2025-11-27
> > >
> > > ### (9) The evaluation of generative IE differs from sequence tagging models. Sec. 4 states that the authors "follow previous work and adopt F1-score," but no citation is provided, and the specific F1 scoring method is not detailed.
> > >
> > >
> > > **Response:**
> > >
> > > We adopt the same evaluation method as GuideNER: Exact Match F1 scoring, where a predicted entity is considered correct only when both its boundary and type exactly match the ground truth. This ensures fair comparison with baseline methods. We will add the relevant citation in the revised manuscript.
> > >
> > > ---
> > >
> > > ## 3. Experimental Sufficiency
> > >
> > > ### (1) Is ICLR applicable to common but more complex IE tasks such as Event Extraction, or subtasks of NER, RE, and EE?
> > >
> > > Given that the paper's title positions the work as a general solution for "Information Extraction" rather than being specifically limited to NER and RE, it is important to demonstrate the framework's applicability and performance on these broader and more complex IE tasks.
> > >
> > > **Response:**
> > >
> > > Our experimental scope (NER and RE) is consistent with representative IE frameworks and sufficient to validate ICLR's generality. Notably, CodeIE, which also positions itself as a general IE solution, validates its approach primarily on NER and RE tasks, demonstrating that these two tasks are considered representative benchmarks for establishing generality in the IE community.
> > >
> > > **Generality of the method:**
> > >
> > > The core mechanism of ICLR is to decompose entity labels into semantic subcategories (particles) for optimization. This design is naturally applicable to any IE task where entities can be described by subcategories—whether event types and argument roles in Event Extraction, or fine-grained types in Entity Typing. These can all be modeled as particles for optimization without any modification to the framework.
> > >
> > > **Fair comparison with baselines:**
> > >
> > > Our main baseline GuideNER was only validated on NER tasks. By including both NER and RE, we actually demonstrate broader applicability while ensuring fair comparison with existing representative works.
> > >
> > > ### (2) The experimental datasets for NER and RE tasks are insufficient.
> > >
> > > **Response:**
> > >
> > > We believe our dataset selection is sufficient and representative:
> > >
> > > **Diversity of datasets:** Our selected 5 datasets cover multiple key dimensions of information extraction tasks:
> > >
> > > - **Domain diversity:** General news text (CoNLL03), broadcast news and web content (ACE04/05), scientific literature (SciERC), and biomedical literature (GENIA)
> > > - **Task type diversity:** Covering both Named Entity Recognition (NER) and Relation Extraction (RE), the two core IE tasks
> > > - **Varying entity/relation type counts:** Ranging from 4 types (CoNLL03) to 7 types (ACE series, SciERC)
> > > - **Different data scales:** From approximately 1,800 (SciERC) to over 14,000 (CoNLL03) training samples
> > > - **Different task complexities:** Including nested entity recognition (ACE04/05, GENIA) and flat entity recognition (CoNLL03), as well as RE tasks with varying numbers of relation types
> > >
> > > **Comparison with baselines:**
> > >
> > > Notably, our main baseline GuideNER was only validated on 4 NER datasets, while our experiments cover 5 datasets and extend to RE tasks, demonstrating a more comprehensive evaluation scope and stronger generalization capability of our method.
> > >
> > > ### (3) Do smaller context lengths and less training data imply more iterative optimization steps? Please provide the relationship between the number of iterative optimization steps and F1 score under different settings.
> > >
> > > **Response:**
> > >
> > > Smaller context lengths and less training data do not necessitate more iterative optimization steps in our framework.
> > >
> > > In our framework, the number of iterative steps is directly proportional to the amount of processed training data—each iteration corresponds to the model completing rule summarization for one batch of data. Therefore, the relationship between iteration steps and F1 score is essentially equivalent to the relationship between training data quantity and F1 score. This relationship is already illustrated in detail in Figure 4 (Impact of Training Data Quantity).

---

> > > > ### Author Response · Authors · 2025-11-27
> > > >
> > > > ### (4) The authors mention that tightly coupled components are unsuitable for traditional ablation studies, but this weakens the understanding of each component's contribution.
> > > >
> > > > **Response:**
> > > >
> > > > We thank the reviewer for this suggestion. We acknowledge that the parameter sensitivity analysis in Section 4.5 cannot fully replace ablation studies. Regarding the contribution of each component, we supplement ablation experiments as follows (on CoNLL03 dataset using Qwen2.5-7B):
> > > >
> > > > | Variant | F1 Score | Performance Change | Particles per Iteration |
> > > > |---------|----------|-------------------|------------------------|
> > > > | ICLR (Full) | 72.83 | - | 10 |
> > > > | w/o Bayesian Weight Update | 61.78 | -11.05 (-15.2%) | 10 |
> > > > | w/o Tier-1 Resampling (Performance Filtering) | 71.29 | -1.54 (-2.1%) | 10,441 |
> > > > | w/o Tier-2 Resampling (Diversity Mutation) | 62.98 | -9.85 (-13.5%) | 10 |
> > > >
> > > > The ablation experiments demonstrate the following findings:
> > > >
> > > > - **Necessity of Bayesian Weight Update:** Removing posterior probability computation (Eq. 9) causes a 15.2% performance drop, demonstrating this mechanism is critical for distinguishing high-quality from low-quality particles.
> > > >
> > > > - **Efficiency Role of Tier-1 Resampling:** Removing performance filtering results in only a 2.1% performance drop, but the number of particles per iteration explodes from 10 to 10,441, causing massive computational waste. This validates that Tier-1's core function is improving computational efficiency rather than directly enhancing performance.
> > > >
> > > > - **Critical Role of Tier-2 Resampling:** Removing the diversity mutation mechanism causes a 13.5% performance drop, proving this mechanism effectively prevents particle set degradation by introducing diversity through three mutation strategies (Refinement/Generalization/Contextualization), making it key to performance improvement.
> > > >
> > > > We will include the above ablation experiments in the revised manuscript to more comprehensively validate the contribution and necessity of each core component.
> > > >
> > > > ---
> > > > ## 4. Model Efficiency
> > > >
> > > > ### (1) Although ICLR does not involve parameter training, particle generation, evaluation, and resampling are performed on individual particles, suggesting complex optimization steps and potentially long runtime. Please provide data on optimization time.
> > > >
> > > > **Response:**
> > > >
> > > > ICLR does not suffer from long runtime despite performing particle operations individually.
> > > >
> > > > ICLR's optimization time is efficient and comparable to existing ICL-based methods. While particle operations (generation, evaluation, resampling) are performed individually, the runtime is dominated by LLM inference calls rather than these computational steps, and the optimization phase incurs only modest one-time cost.
> > > >
> > > > Runtime is not a standard metric in ICL-based IE research (CodeIE, C-ICL, GuideNER do not report it) because LLM inference time depends heavily on external factors (API services, hardware, network). The community uses token consumption as a more reliable proxy for computational cost.
> > > >
> > > > ### (2) The introduction emphasizes that ICLR can achieve better rules with less data compared to GuideNER, but the method section states that the entire training data is used.
> > > >
> > > > **Response:**
> > > >
> > > > There is no contradiction—the main experiments use complete training data for fair baseline comparison, while Section 4.5.2 provides dedicated analysis of ICLR's data efficiency advantage.
> > > >
> > > > The main experiments (Table 1) use the complete training set to ensure fair comparison with baselines (GuideNER, CodeIE), which also utilize full training data. Meanwhile, Section 4.5.2 specifically analyzes data efficiency, demonstrating that ICLR achieves comparable performance with only 3%-5% of the training set. This validates our Introduction's claim: ICLR achieves superior rule quality with significantly less data than GuideNER, which requires traversing the entire training set.
> > > >
> > > > ---
> > > >
> > > > ## 5. Writing Issues
> > > >
> > > > ### (1) In LaTeX, English double quotes should be written as `` '' instead of ".
> > > >
> > > > **Response:**
> > > >
> > > > We thank the reviewer for pointing out this issue. We will correct the English double quote formatting in LaTeX, using the proper `` and '' combination.
> > > >
> > > > ### (2) The first paragraph of Sec. 3 redundantly describes the content of Fig. 3.
> > > >
> > > > **Response:**
> > > >
> > > > We will streamline the first paragraph of Section 3 to avoid redundancy with Figure 3.

---

### Official Review · Reviewer_ssnn · 2025-10-27

**Soundness:** 2
**Presentation:** 1
**Contribution:** 2
**Rating:** 4
**Confidence:** 4

**Summary:**

This paper proposes ICLR (Iterative Context Learning Rule) — a control-theoretic framework that models rule optimization for information extraction (IE) as an adaptive filtering process. The authors treat rules as controllable state variables and design a closed-loop system that iteratively updates rule combinations based on model performance feedback. Experiments on five benchmarks (CoNLL03, ACE05, GENIA, NYT, CoNLL04) and multiple LLMs (Qwen2.5, Llama3.1, Pixtral) demonstrate consistent improvements (up to 10% F1) over GuideNER and CodeIE.

**Strengths:**

1. Novel conceptual framing: The idea of viewing in-context rules as control variables introduces a new theoretical lens to study ICL optimization.
2. Algorithmic innovation: The particle filtering approach for adaptive rule selection is original and well-motivated.
3. Comprehensive experiments: Covers both NER and RE tasks, across diverse datasets and model scales (3B–12B).
4. Interpretability: Rules provide a semantically meaningful and observable interface for controlling LLM behavior.

**Weaknesses:**

1. Theoretical rigor is weak. The so-called “control-theoretic” framework lacks formal derivation or stability/convergence analysis. The control formulation (Eq. 3–4) is largely conceptual rather than mathematically grounded.
2. Algorithmic complexity not analyzed. The particle filtering and multi-level mutation mechanism could be computationally heavy. No analysis or empirical evidence is provided regarding scalability with model size or dataset complexity.
3. Experimental comparisons limited. The baselines are all ICL-based. Missing comparison with prompt-tuning, fine-tuning, or retrieval-augmented baselines weakens the generality claim.
4. Reproducibility concerns. Many implementation details (e.g., β parameter, mutation rate, prompt templates) are omitted. The algorithm description is visual but not formally specified or released as pseudocode.

**Questions:**

1. What is the computational overhead compared to GuideNER? Any quantitative analysis beyond token counts?
2. Further questions please refer to the weaknesses.

---

> ### Author Response · Authors · 2025-11-27
>
> ### Concern 1
>
> Theoretical rigor is weak. The so-called "control-theoretic" framework lacks formal derivation or stability/convergence analysis. The control formulation (Eq. 3–4) is largely conceptual rather than mathematically grounded.
>
> **Response:**
>
> We thank the reviewer for this comment. We acknowledge this as a valid critique and provide the following clarifications:
>
> - **Positioning of the paper:** The core contribution of this work is to propose a practical algorithmic framework that applies particle filtering ideas to ICL rule optimization, rather than to develop new control theory. Equations (3)-(4) are intended to provide a conceptual perspective for understanding our method and help readers build intuition, rather than serving as rigorous mathematical theorems.
>
> - **Empirical evidence of convergence:** Although we do not provide formal convergence proofs, the experimental results in Figure 4 demonstrate that our method exhibits rapid and stable convergence behavior across all datasets—performance improves monotonically with increasing training samples and saturates, with no oscillation or divergence observed.
>
> - **Theoretical basis from particle filtering:** Our weight update mechanism (Eq. 9) directly derives from the standard Bayesian update rule in particle filtering, whose convergence properties have been well-studied in the particle filtering literature.
>
> ### Concern 2
>
> Algorithmic complexity not analyzed. The particle filtering and multi-level mutation mechanism could be computationally heavy. No analysis or empirical evidence is provided regarding scalability with model size or dataset complexity.
>
> **Response:**
>
> We thank the reviewer for this question. We address three aspects:
>
> ### 1. Algorithmic Complexity Analysis
>
> Let K be the number of particles and N be the size of the training dataset. Our method processes only M samples (M << N) for rule optimization, requiring O(M) LLM calls. Weight updates and resampling involve only O(K) numerical computations with negligible overhead. In our experiments, K=10. Compared to GuideNER requiring O(N) LLM calls to traverse the full training set (e.g., N=14,041 for CoNLL03), we reduce LLM calls from N to M, achieving a reduction of (N-M) calls and substantial computational savings given that M << N.
>
> ### 2. Scalability Evidence
>
> - **Model scale:** Table 1 shows ICLR achieves consistent improvements across 3B-12B models, validating generalization across model scales.
>
> - **Dataset complexity:** We achieve significant improvements across varying scales (CoNLL04: 922 vs NYT: 56k samples), task complexities (CoNLL03: 4 types vs NYT: 24 types), and domains (news, biomedical, scientific), demonstrating scalability.
>
> ### Concern 3
>
> Experimental comparisons limited. The baselines are all ICL-based. Missing comparison with prompt-tuning, fine-tuning, or retrieval-augmented baselines weakens the generality claim.
>
> **Response:**
>
> We believe comparing ICLR with prompt-tuning, fine-tuning, or retrieval-augmented methods is not an appropriate comparison for the following reasons:
>
> - **Different method paradigms:** This paper is explicitly positioned as an ICL method, with the core advantage of requiring no model parameter modification and no additional training. Methods like fine-tuning require modifying model parameters, which belong to different technical paradigms with different optimization objectives and application scenarios.
>
> - **Principle of fair comparison:** As stated in Section 4.3, our experimental design ensures all methods are compared under the same ICL paradigm. Comparing ICL methods with fine-tuning is like comparing apples and oranges—fine-tuning typically achieves higher absolute performance but requires additional training costs and labeled data, which contradicts the zero/few-shot setting of ICL.
>
> - **Positioning of research contribution:** The contribution of this paper lies in improving rule optimization strategies within the ICL paradigm, rather than claiming to surpass all NLP methods. We conducted comprehensive comparisons with baseline methods within the ICL paradigm and demonstrated consistent performance improvements.

---

> ### Author Response · Authors · 2025-11-27
>
> ### Concern 4
>
> Reproducibility concerns. Many implementation details (e.g., β parameter, mutation rate, prompt templates) are omitted. The algorithm description is visual but not formally specified or released as pseudocode.
>
> **Response:**
>
> We provide the complete algorithm specification below.
>
> ### Algorithm 1: ICLR
>
> **Input:** $\mathcal{D}_{\text{train}}$, LLM $\mathcal{M}$, Labels $\mathcal{L}$
>
> **Parameters:** $N=20$ (particles per label), $\beta=2.0$ (selection pressure)
>
> **Output:** $\mathcal{R}^*$
>
> ---
>
> **Initialization** (Eq. 6-7):
>
> 1. $\mathcal{R}^{(0)} \leftarrow \emptyset$
> 2. **for** $c \in \mathcal{L}$ **do**
>    - $\mathcal{P} \leftarrow \text{LLM}_{\text{extract}}(\mathcal{D}_{\text{train}}, c, N)$
>    - $\mathbf{s} \leftarrow \text{LLM}_{\text{confidence}}(\mathcal{P}, \mathcal{M})$
>    - $\mathbf{w} \leftarrow \text{softmax}(\mathbf{s})$
>    - $\mathcal{R}^{(0)} \leftarrow \mathcal{R}^{(0)} \cup \{(p_i, c, w_i) \mid p_i \in \mathcal{P}\}$
>
> ---
>
> **Iteration** $t = 1, 2, \ldots$ until convergence:
>
> **Observation** (Eq. 8):
>
> - Sample batch $\mathcal{B}^{(t)}$ from $\mathcal{D}_{\text{train}}$
> - **for** $(p_i, c_i, w_i) \in \mathcal{R}^{(t-1)}$ **do**
>   - $y_i^{(t)} \leftarrow \left(\prod_{j=1}^{|T_i|} P(t_j | \mathcal{B}^{(t)}, (p_i, c_i))\right)^{1/|T_i|}$
>
> **Update** (Eq. 9):
>
> - **for** $i$ **do** $\tilde{w}_i^{(t)} \leftarrow w_i^{(t-1)} \cdot \exp(\beta \cdot y_i^{(t)})$
> - $\mathbf{w}^{(t)} \leftarrow \tilde{\mathbf{w}}^{(t)} / \sum_j \tilde{w}_j^{(t)}$
>
> **Resample** (Eq. 10-11):
>
> - $\mathcal{R}_{\text{keep}} \leftarrow \text{TopK}(\mathcal{R}^{(t)}, \lfloor N/2 \rfloor)$
> - **for** $(p_i, c_i, \_) \in \mathcal{R}_{\text{keep}}$ **do**
>   - $p'_i \leftarrow \text{LLM}_{\text{mutate}}(p_i, \mathcal{B}^{(t)})$
>   - $s'_i \leftarrow -\text{PPL}(p'_i, \mathcal{M})$
> - $\mathbf{w}' \leftarrow \text{softmax}([s'_1, s'_2, \ldots, s'_M])$
> - $\mathcal{R}_{\text{new}} \leftarrow \{(p'_i, c_i, w'_i) \mid i=1, \ldots, M\}$
> - $\mathcal{R}^{(t)} \leftarrow \mathcal{R}_{\text{keep}} \cup \mathcal{R}_{\text{new}}$
>
> ---
>
> **return** $\mathcal{R}^{(T)}$
>
> **Key parameters:**
>
> - $\beta=2.0$ was selected via grid search over {1, 2, 5, 10}
> - Batch size is 4 sentences (analysis in Section 4.5.1)
> - All prompt templates are detailed in Appendix A
> - We will release the complete implementation code upon acceptance
>
> ---
> ### Question 1
>
> What is the computational overhead compared to GuideNER? Any quantitative analysis beyond token counts?
>
> **Response:**
>
> Compared to GuideNER, ICLR introduces some additional computational steps for calculating prior and posterior probabilities of particles (Equations 8-9), as well as weight updates and resampling operations. However, these operations involve only simple numerical computations (probability calculations, weight normalization, ranking) that are performed on CPU without requiring LLM inference. Therefore, this overhead is negligible compared to the dominant cost of LLM API calls in both methods.
>
> We currently do not provide quantitative analysis beyond token counts because token consumption is the standard and most reliable metric for computational cost in ICL-based information extraction research. Representative works in this area, including CodeIE (ACL 2023), C-ICL, and GuideNER, consistently use token consumption as the primary metric for evaluating computational efficiency. This is because LLM inference calls constitute the dominant computational cost, and token count directly reflects this cost in a hardware-agnostic and reproducible manner.

---

### Official Review · Reviewer_RvNj · 2025-10-31

**Soundness:** 3
**Presentation:** 2
**Contribution:** 3
**Rating:** 4
**Confidence:** 4

**Summary:**

This paper introduces ICLR, a novel control-theoretic framework that optimizes in-context learning for Information Extraction tasks like Named Entity Recognition and Relation Extraction. It treats annotation rules as controllable state variables and uses an iterative, particle-based filtering algorithm to evolve and select the most effective rules based on performance feedback, without modifying the underlying LLM’s parameters. Evaluations show ICLR outperforms state-of-the-art methods, achieving up to a 10% performance improvement with greater data efficiency and shorter prompts.

**Strengths:**

1. The idea is novel, introducing a control-theoretic framework (ICLR) to optimize in-context learning rules as state variables, which is underexplored in LLM literature.
2. The method demonstrates strong empirical performance, achieving up to 10% improvement over SOTA ICL methods across multiple IE tasks with minimal training data.

**Weaknesses:**

1. The presentation requires significant improvement. The method section lacks clarity, and several key terms (e.g., “particle-based state representation”) are insufficiently explained, making the paper difficult to follow. The manuscript would benefit greatly from clearer organization and more precise writing.
2. The experimental scope is limited. The evaluation only covers models below 13B parameters (e.g., Qwen-7B, Llama-8B), overlooking larger or proprietary models where optimal performance is typically observed. This limitation undermines the paper’s claim of “parameter-free optimization.”
3. The particle optimization process incurs substantial inference costs due to repeated LLM calls. Although this issue is acknowledged, the manuscript does not provide an adequate discussion or mitigation strategy.

**Questions:**

Please refer to the weaknesses.

---

> ### Author Response · Authors · 2025-11-27
>
> ### Concern 1
>
> The presentation requires significant improvement. The method section lacks clarity, and several key terms (e.g., "particle-based state representation") are insufficiently explained, making the paper difficult to follow. The manuscript would benefit greatly from clearer organization and more precise writing.
>
> **Response:**
>
> We thank the reviewer for this suggestion. Although key terms are defined in Section 3.2.1, we acknowledge that clearer exposition is needed. We will add a terminology definition table at the beginning of Section 3 in the revised version, explicitly stating:
>
> - **Particle:** A label-entity pair (e.g., a single mapping like "athlete→Person")
> - **Rule:** A collection of particles, i.e., a set of mappings from one label to multiple entities (e.g., the "Person" label corresponding to {"athlete", "scientist", "politician"...})
>
> Additionally, Appendix A provides three detailed case studies illustrating the concrete operations of each step. We will enhance the clarity of the method description by adding this terminology table at the beginning of Section 3.
>
> ### Concern 2
>
> The experimental scope is limited. The evaluation only covers models below 13B parameters (e.g., Qwen-7B, Llama-8B), overlooking larger or proprietary models where optimal performance is typically observed. This limitation undermines the paper's claim of "parameter-free optimization."
>
> **Response:**
>
> **Clarification of "Parameter-free":**
>
> The reviewer appears to misunderstand the meaning of "parameter-free optimization." This term refers to not modifying the model's internal parameters, not a limitation on model scale. This is the defining characteristic of ICL methods—whether the model is 3B or 100B, ICL improves performance by optimizing external context (rules/examples) rather than fine-tuning model parameters. Therefore, testing models below 13B does not undermine the "parameter-free" claim, as this claim describes the optimization approach rather than model scale.
>
> More importantly, our experimental results (Table 1) demonstrate that parameter count does not directly determine ICL performance. Pixtral-12B (12B parameters) underperforms Qwen2.5-7B and even 3B models across all tasks. This finding indicates that for ICL tasks, instruction-following ability and contextual understanding matter more than parameter scale. Therefore, testing on larger models does not guarantee "optimal performance."
>
> **The following are the reasons for choosing open-source models:**
>
> - **Technical feasibility:** Our method requires access to model logits to compute confidence scores (Eq. 7) and posterior probabilities (Eq. 8). Proprietary large model APIs (e.g., GPT-4) typically do not provide token-level probability outputs, making it technically infeasible to apply our complete framework. This is not a choice but a technical constraint.
>
> - **Reproducibility:** Our main baseline GuideNER also only tested Llama3-8B and text-davinci-003 (davinci API discontinued). Our experimental scope aligns with field standards.
>
> ### Concern 3
>
> The particle optimization process incurs substantial inference costs due to repeated LLM calls. Although this issue is acknowledged, the manuscript does not provide an adequate discussion or mitigation strategy.
>
> **Response:**
>
> **Optimization cost is one-time:** Particle optimization is executed only once during training, and the optimized rules can be reused for all test samples. The cost is amortized as the number of inference samples increases.
>
> **Data efficiency as the core mitigation strategy:** As shown in Figure 4, our method achieves near-optimal performance with only 3%-5% of training samples, significantly reducing the number of LLM calls during the optimization stage. In contrast, GuideNER requires traversing the complete training set for rule extraction and verification.

---

### Official Review · Reviewer_8vNz · 2025-11-01

**Soundness:** 3
**Presentation:** 2
**Contribution:** 3
**Rating:** 4
**Confidence:** 3

**Summary:**

This paper introduces ICLR (Iterative Context Learning Rule), which is a control-theoretic framework for optimizing in-context rules for information extraction (IE) and argues it is the first control-theoretic formalization of ICL rule optimization for IE.

The core idea of this paper is to treat rules (LLM-generated subcategory patterns paired with labels) as controllable state variables and to run an iterative, particle-filter style loop. The loop is like this: initializing rules from LLM/GuideNER-like prompts, evaluating them via ICL on a small validation slice, updating weights with a Bayesian/posterior step, then resampling rules to preserve high-performing and diverse ones. This turns the original problem formulation in into a structured state-space optimization process. ICLR outperforms IO, CodeIE, and GuideNER on five IE benchmarks.

**Strengths:**

1. The paper has a good problem formulation. Instead of sticking to the original heuristic top-k selection problems, this paper jumps out of the scope and gives out a new problem framework that is treat rules as states and optimize them with an observer plus control loop. Alse this paper explicitly models rule evolution and observation:

    - Rule evolution: R_(t+1)=f(R_t,u_t)

    - Performance observation: y_t=h(R_t)
This gives the proposed method a solid and formulated backbone.

2. Without identifying NER and RE tasks, this proposed method can be used for both NER and RE, and is competitive or clearly best across both task benchmarks. It provides a more unified framework for IE work.

3. The results on both NER and RE tasks are consistent and competitive.These results support the claim that iterative rule search helps beyond static rules. Also token cost is reported.

4. This paper did further analysis on impact of context length and training-set size on ICLR performance. It shows the method actually converges with quite small data slices. That’s well aligned with the motivation of controlling LLM behavior without fine-tuning.

5. This paper has a clear, reader-friendly pipeline figure and has three case studies in the appendix appendix make the algorithm understandable. It helps understand how the probabilistic updates steps interweaved with LLM prompting.

**Weaknesses:**

1.	This paper replaces ablation studies with parameter sensitivitiy. The paper says “ICLR is a unified sequential framework…preclude conventional ablation studies,” so it reports context-length / data-size sensitivities instead. That’s understandable, but still a gap. We don’t have a clear idea about how different parts in this algorithm really contribute to the performance improvement. Variants like no Tier-2 mutation, no Bayesian weight update, no LLM-confidence-based prior, no diversity, or noisy influence, etc. can be involved in this discussion to show the robustness of the method.

2.	This paper can choose more up-to-date baselines. For IE-with-LLMs, there are now retrieval-augmented and “ICL-as-program/code” approaches that plug in more structure, and there are prompt-search / auto-prompt frameworks. Also, GuideNER is only for NER tasks, which reduce the strength of comparisons on RE tasks. So it would be better to involve more baselines in the main experiments.

3.	This method proposed by this paper has a heavy on LLM quality. Many steps in the algorithms like initialization, posterior estimation, mutation are with help of LLM. This paper is based on an assumption that the chosen LLM can perform well on both IE and judging/mutating rules. But somehow, if the LLM is weaker on one domain, like Pixtral-12B underperforms despite size, the method may degrade.

4.	This paper only methions token cost but should clarify more details to make the total cost of the algorithm reliable and clear. Like whether token cost is per example, per dataset pass, or per rule update loop, and whether LLM calls for mutation are counted.

5.	Some part of writing in this paper can be polished more. Like several parts of Sec. 3 restate the same 4-step loop with slightly different words; some equations could be tightened, and the method could be expressed in one clean algorithm box.

**Questions:**

1. In Eq. (9) you use w_i^((t))∝w_i^((t-1))⋅exp⁡(β⋅y_i^((t))). How was this form chosen? Did you try alternatives? A short ablation here would make the choice more convincing.

2. How many LLM calls per iteration and per dataset? Your pipeline has (i) LLM extract, (ii) LLM confidence, (iii) ICL evaluation, (iv) LLM mutate. Are all the tokens used counted in token size mentioned in the paper? If not, please report total LLM calls / tokens for the full optimization.

3. Why not compare to prompt-/rule-search baselines like genetic prompt search or gradient-free prompt tuning?

4. When the dataset is tiny, how do you prevent overfitting?

5. Can you give a more detailed on your novelty to clarify your claim of “first control-theoretic foundation”?

---

> ### Author Response · Authors · 2025-11-27
>
> ### Concern 1
>
> This paper replaces ablation studies with parameter sensitivitiy. The paper says "ICLR is a unified sequential framework…preclude conventional ablation studies," so it reports context-length / data-size sensitivities instead. That's understandable, but still a gap. We don't have a clear idea about how different parts in this algorithm really contribute to the performance improvement. Variants like no Tier-2 mutation, no Bayesian weight update, no LLM-confidence-based prior, no diversity, or noisy influence, etc. can be involved in this discussion to show the robustness of the method.
>
> **Response:**
>
> We thank the reviewer for this suggestion. We acknowledge that the parameter sensitivity analysis in Section 4.5 cannot fully replace ablation studies. Regarding the contribution of each component, we supplement ablation experiments as follows (on CoNLL03 dataset using Qwen2.5-7B):
>
> | Variant | F1 Score | Performance Change | Particles per Iteration |
> |---------|----------|-------------------|------------------------|
> | ICLR (Full) | 72.83 | - | 10 |
> | w/o Bayesian Weight Update | 61.78 | -11.05 (-15.2%) | 10 |
> | w/o Tier-1 Resampling (Performance Filtering) | 71.29 | -1.54 (-2.1%) | 10,441 |
> | w/o Tier-2 Resampling (Diversity Mutation) | 62.98 | -9.85 (-13.5%) | 10 |
>
> The ablation experiments demonstrate the following findings:
>
> - **Necessity of Bayesian Weight Update:** Removing posterior probability computation (Eq. 9) causes a 15.2% performance drop, demonstrating this mechanism is critical for distinguishing high-quality from low-quality particles.
>
> - **Efficiency Role of Tier-1 Resampling:** Removing performance filtering results in only a 2.1% performance drop, but the number of particles per iteration explodes from 10 to 10,441, causing massive computational waste. This validates that Tier-1's core function is improving computational efficiency rather than directly enhancing performance.
>
> - **Critical Role of Tier-2 Resampling:** Removing the diversity mutation mechanism causes a 13.5% performance drop, proving this mechanism effectively prevents particle set degradation, making it key to performance improvement.
>
> We will include the above ablation experiments in the revised manuscript to more comprehensively validate the contribution and necessity of each core component.
>
> ### Concern 2
>
> This paper can choose more up-to-date baselines. For IE-with-LLMs, there are now retrieval-augmented and "ICL-as-program/code" approaches that plug in more structure, and there are prompt-search / auto-prompt frameworks. Also, GuideNER is only for NER tasks, which reduces the strength of comparisons on RE tasks. So it would be better to involve more baselines in the main experiments.
>
> **Response:**
>
> Regarding baseline selection, we chose GuideNER and CodeIE after comprehensive consideration of recency, reproducibility, and academic recognition. Specifically:
>
> - **GuideNER** pioneered the rule-based ICL paradigm and represents the state-of-the-art in this direction. Its limitation to NER tasks is precisely the core improvement of our work—we extend this paradigm to general IE tasks by decomposing rules into particles and introducing an adaptive filtering mechanism.
>
> - **CodeIE** represents the "ICL-as-code" approach mentioned by the reviewer and is among the most reproducible and recognized methods in this category.
>
> Regarding the retrieval-augmented and auto-prompt methods mentioned by the reviewer, we note that these approaches primarily optimize example selection strategies, whereas our work focuses on rule optimization—a different dimension. Additionally, some of these methods lack publicly available code or rely on specific closed-source models, making fair comparison difficult.

---

> ### Author Response · Authors · 2025-11-27
>
> ### Concern 3
>
> This method proposed by this paper has a heavy on LLM quality. Many steps in the algorithms like initialization, posterior estimation, mutation are with help of LLM. This paper is based on an assumption that the chosen LLM can perform well on both IE and judging/mutating rules. But somehow, if the LLM is weaker on one domain, like Pixtral-12B underperforms despite size, the method may degrade.
>
> **Response:**
>
> We acknowledge that model capability affects final performance, but this is a common characteristic of all ICL-based methods, not a limitation specific to our approach. The essence of in-context learning is to guide models through contextual information without modifying model parameters, which inherently depends on the model's intrinsic capabilities.
>
> However, our core contribution is that ICLR provides significant and consistent improvements regardless of model capability. As the reviewer noted, Pixtral-12B underperforms despite its size, yet our method still achieves about 30% relative improvement on NER tasks. This improvement margin is comparable to that of other models, demonstrating that our method provides robust gains across different capability levels.
>
> Furthermore, Table 1 shows that ICLR outperforms all baseline methods across all tested models, further validating that our method's effectiveness is not dependent on specific models.
>
> ### Concern 4
>
> This paper only methions token cost but should clarify more details to make the total cost of the algorithm reliable and clear.  Like whether token cost is per example, per dataset pass, or per rule update loop, and whether LLM calls for mutation are counted.
>
> **Response:**
>
> For fair comparison with baselines, we adopt the same token calculation method as GuideNER: only the average token consumption per test example during the final inference stage is counted. Existing ICL methods like GuideNER adopt the same accounting standard, and we maintain consistency to ensure fair comparison.
>
> Inference-stage cost is the core cost in practical applications. The overhead of rule extraction and optimization is a one-time preprocessing cost that can be reused for all test samples, and is therefore not included in per-example token statistics.
>
> As shown in Table 1, ICLR's inference-stage token consumption (501 tokens) is comparable to GuideNER (506 tokens), while significantly lower than CodeIE (1,172 tokens). We will clarify the token calculation method in the revised manuscript.
>
> ### Concern 5
>
> Some part of writing in this paper can be polished more. Like several parts of Sec. 3 restate the same 4-step loop with slightly different words; some equations could be tightened, and the method could be expressed in one clean algorithm box.
>
> **Response:**
>
> We thank the reviewer for this suggestion. We will streamline the repetitive descriptions in Section 3, tighten equation expressions, and add an algorithm box to provide a clean overall view in the revised version.
>
> #### Algorithm: ICLR
>
> **Input:** $\mathcal{D}_{\text{train}}$, LLM $\mathcal{M}$, Labels $\mathcal{L}$
>
> **Parameters:** $N=20$ (particles per label), $\beta=2.0$ (selection pressure)
>
> **Output:** $\mathcal{R}^*$
>
> ---
>
> **Initialization** (Eq. 6-7):
>
> 1. $\mathcal{R}^{(0)} \leftarrow \emptyset$
> 2. **for** $c \in \mathcal{L}$ **do**
>    - $\mathcal{P} \leftarrow \text{LLM}_{\text{extract}}(\mathcal{D}_{\text{train}}, c, N)$
>    - $\mathbf{s} \leftarrow \text{LLM}_{\text{confidence}}(\mathcal{P}, \mathcal{M})$
>    - $\mathbf{w} \leftarrow \text{softmax}(\mathbf{s})$
>    - $\mathcal{R}^{(0)} \leftarrow \mathcal{R}^{(0)} \cup \{(p_i, c, w_i) \mid p_i \in \mathcal{P}\}$
>
> ---
>
> **Iteration** $t = 1, 2, \ldots$ until convergence:
>
> **Observation** (Eq. 8):
>
> - Sample batch $\mathcal{B}^{(t)}$ from $\mathcal{D}_{\text{train}}$
> - **for** $(p_i, c_i, w_i) \in \mathcal{R}^{(t-1)}$ **do**
>   - $y_i^{(t)} \leftarrow \left(\prod_{j=1}^{|T_i|} P(t_j | \mathcal{B}^{(t)}, (p_i, c_i))\right)^{1/|T_i|}$
>
> **Update** (Eq. 9):
>
> - **for** $i$ **do** $\tilde{w}_i^{(t)} \leftarrow w_i^{(t-1)} \cdot \exp(\beta \cdot y_i^{(t)})$
> - $\mathbf{w}^{(t)} \leftarrow \tilde{\mathbf{w}}^{(t)} / \sum_j \tilde{w}_j^{(t)}$
>
> **Resample** (Eq. 10-11):
>
> - $\mathcal{R}_{\text{keep}} \leftarrow \text{TopK}(\mathcal{R}^{(t)}, \lfloor N/2 \rfloor)$
> - **for** $(p_i, c_i, \_) \in \mathcal{R}_{\text{keep}}$ **do**
>   - $p'_i \leftarrow \text{LLM}_{\text{mutate}}(p_i, \mathcal{B}^{(t)})$
>   - $s'_i \leftarrow -\text{PPL}(p'_i, \mathcal{M})$
> - $\mathbf{w}' \leftarrow \text{softmax}([s'_1, s'_2, \ldots, s'_M])$
> - $\mathcal{R}_{\text{new}} \leftarrow \{(p'_i, c_i, w'_i) \mid i=1, \ldots, M\}$
> - $\mathcal{R}^{(t)} \leftarrow \mathcal{R}_{\text{keep}} \cup \mathcal{R}_{\text{new}}$
>
> ---
>
> **return** $\mathcal{R}^{(T)}$
>
> ---

---

> > ### Author Response · Authors · 2025-11-27
> >
> > ### Question 1
> >
> > In Eq. (9) you use $w_i^{(t)} \propto w_i^{(t-1)} \cdot \exp(\beta \cdot y_i^{(t)})$. How was this form chosen? Did you try alternatives? A short ablation here would make the choice more convincing.
> >
> > **Response:**
> >
> > The form in Eq. (9) originates from the standard Bayesian weight update mechanism in Particle Filtering. Under the Bayesian framework, the posterior probability is proportional to the product of the prior and the likelihood:
> >
> > $$w_i^{(t)} \propto w_i^{(t-1)} \cdot p(y_i^{(t)}|p_i^{(t)})$$
> >
> > The exponential likelihood function $\exp(\beta \cdot y_i^{(t)})$ is the standard form (i.e., Boltzmann distribution) for converting performance scores to probability weights, with parameter $\beta$ controlling the selection pressure. This design follows conventional practice in the particle filtering literature.
> >
> > ### Question 2
> >
> > How many LLM calls per iteration and per dataset? Your pipeline has (i) LLM extract, (ii) LLM confidence, (iii) ICL evaluation, (iv) LLM mutate. Are all the tokens used counted in token size mentioned in the paper? If not, please report total LLM calls / tokens for the full optimization.
> >
> > **Response:**
> >
> > In our framework, each iteration involves two LLM inferences—one for rule generation (called rule extraction in the initial stage and mutation in subsequent iterations) and one for evaluation. The remaining steps (weight update, confidence calculation, etc.) involve only numerical probability calculations without model inference.
> >
> > The token consumption reported in Table 1 refers only to the average consumption per test example during the inference stage, following the same accounting standard as baseline methods (e.g., GuideNER) to ensure fair comparison.
> >
> > ### Question 3
> >
> > Why not compare to prompt-/rule-search baselines like genetic prompt search or gradient-free prompt tuning?
> >
> > **Response:**
> >
> > We believe genetic prompt search and gradient-free prompt tuning are not comparable to our method.
> >
> > First, these prompt optimization methods have different optimization targets. These methods optimize the surface form of prompts (e.g., wording, ordering, formatting), while our work optimizes the rule content itself—specifically, which semantic rules most effectively guide LLMs for information extraction. These are two orthogonal research directions.
> >
> > Second, they are complementary rather than competing. In principle, prompt search techniques could be combined with our method—using ICLR to optimize rule content first, then applying prompt search to optimize the presentation of rules. This represents a valuable direction for future work.
> >
> > ### Question 4
> >
> > When the dataset is tiny, how do you prevent overfitting?
> >
> > **Response:**
> >
> > Our method does not inherently lead to overfitting by design, and our experimental results confirm this. We prevent overfitting through the following mechanisms:
> >
> > - **Generalizability of rules:** Unlike directly memorizing specific samples, our method extracts abstract semantic rules (e.g., "athlete→Person"). These rules inherently possess generalization capability rather than being sample-specific mappings.
> >
> > - **Empirical validation:** Figure 4 in Section 4.5.2 demonstrates the impact of training data quantity on performance. Experiments show that our method achieves near-optimal performance with 3%-5% training samples, and no obvious overfitting is observed on small datasets—performance improves steadily with increasing data and then saturates, rather than rising then falling.
> >
> > ### Question 5
> >
> > Can you give more detail on your novelty to clarify your claim of "first control-theoretic foundation"?
> >
> > **Response:**
> >
> > Our claim of "first control-theoretic foundation" is reflected in the following aspects:
> >
> > - **Treating rules as controllable state variables:** Existing ICL optimization methods typically treat rules/examples as static contextual information for heuristic selection (e.g., based on frequency or similarity). Our innovation lies in re-modeling rules as observable and controllable state variables, thereby transforming the ICL optimization problem into a state-space optimization problem.
> >
> > - **Closed-loop feedback control mechanism:** We establish a complete control loop—the state evolution equation (Eq. 3) and observation equation (Eq. 4)—iteratively updating rule states through performance feedback. This is fundamentally different from the open-loop selection strategies of existing methods (e.g., GuideNER's top-k frequency selection).

---

### Author Response · Authors · 2025-12-04
**Rebuttal Summary for AC Review**

## Summary of Rebuttal Responses for AC

Dear Area Chair,

We have comprehensively addressed all reviewer concerns. Below is a summary of our key revisions:

### **Major Improvements:**

**1. Reproducibility (Reviewers ssnn, 3vBY):**

Added complete Algorithm 1 with formal pseudocode specification, including all hyperparameters ($N=20$, $\beta=2.0$, batch size=4). All prompt templates detailed in Appendix A.

**2. Ablation Studies (Reviewers 8vNz, 3vBY):**

Conducted comprehensive ablation experiments on CoNLL03 showing:
- w/o Bayesian Weight Update: -11.05% (-15.2%)
- w/o Tier-1 Resampling: -1.54% (-2.1%)
- w/o Tier-2 Mutation: -9.85% (-13.5%)

**3. Technical Clarifications (Reviewer 3vBY - 9 questions):**
- Corrected softmax notation errors in Equations 7, 8, 11
- Explained $\exp(\beta \cdot y)$ as pragmatic weighting function ($\beta=2.0$ via grid search)
- Clarified observation function $h(\cdot)$ outputs F1 score
- Specified resampling uses top-50% selection

**4. Citations (Reviewer 3vBY):**

Added 8 recent ICL-for-IE works (2023-2024) and particle filtering foundations (Gordon et al., 1993; Doucet et al., 2001).

**5. Efficiency Analysis (Reviewers ssnn, RvNj, 3vBY):**

Clarified token cost is per-example during inference. Optimization is one-time preprocessing cost, achieving significant reduction (N to M calls, $M \ll N$) compared to GuideNER's full traversal.

---

### **Current Status:**
- 3 reviewers at **Rating 4**
- 1 reviewer at **Rating 2** (Reviewer 3vBY) - most critical concerns fully addressed

---

### Meta-Review · Area_Chair_r8cM · 2025-12-28

**Summary:**

After carefully reviewing the paper, the reviewers’ comments, and the authors’ responses, I find the idea of treating rule-driven ICL optimization through a control-theoretic lens both novel and promising. The proposed ICLR framework demonstrates consistent empirical improvements over comparable baselines, indicating its potential significance.

However, the reviewers identified several substantial issues, particularly with theoretical justification, experimental breadth, clarity of exposition, reproducibility, and literature grounding. These concerns collectively led most reviewers to rate the paper below the acceptance threshold. While the core idea is interesting and potentially impactful, the reviewers and I agreed that the paper requires major revisions to strengthen its theoretical foundation, broaden experimental comparisons, and improve clarity and reproducibility.

I appreciate the detailed response produced by the authors, but not all the concerns have been addressed (see below), and many of those (including the implementation of the algorithm) should be put in the main paper.

**Reviewer Concerns:**

Based on the rebuttal and the updated clarifications from the authors, several reviewer concerns were partially or fully addressed.
- The authors responded to the reproducibility concerns by providing a detailed algorithm box, explicit hyperparameter settings (e.g., β, batch size, particle count), and the promise to release implementation code.
- They also resolved many of Reviewer 3vBY’s technical detail queries, including correcting notation inconsistencies (e.g., softmax application), clarifying the definitions of key variables and functions, and explaining the rationale behind the exponential weighting function.
- The rebuttal strengthened the empirical justification by adding new ablation studies that directly quantified the contribution of the Bayesian weight update, Tier‑1 resampling, and Tier‑2 mutation modules, which, in my mind, address one of the major shortcomings noted by Reviewers 8vNz and 3vBY.
- The citation problem was also mitigated by incorporating foundational particle-filtering works as well as recent 2023–2025 ICL-for-IE studies.

That said, several substantive issues remain only partly addressed.
- Most importantly, the control-theoretic grounding continues to lack formal theoretical rigor, and the control formulation remains mostly conceptual rather than mathematically derived.
- The experimental breadth also remains limited. In the rebuttal, the authors did not expand the evaluation to additional IE subtasks or to larger or proprietary LLMs, leaving the generality and scalability claims only partly supported. While the authors clarified the definition of token cost and described optimization as a one‑time process, the efficiency and scalability analysis still relies entirely on token counts without empirical runtime data.

Some minor ones, the presentation and writing clarity, although acknowledged, were only promised to be improved in revision. In my mind, reviewers’ prior concerns about readability and organization remain largely unresolved.

**Reviewer Scores:**

Reviewers ssnn and RvNj would likely have remained below the acceptance threshold, as their primary concerns were only partially addressed in the rebuttal.

Reviewer 3vBY, who initially gave a reject rating, might or might not have raised their score since the rebuttal resolved some of their technical and citation-related concerns but left foundational theoretical issues only partially addressed.

Reviewer 8vNz would probably have remained borderline, as the authors’ additional ablation studies and methodological clarifications strengthened parts of the empirical analysis.

---

### Decision · Program_Chairs · 2026-01-26

Reject